# Better Replacement for TTS Naturalness Evaluation

*Sajad Shirali-Shahreza*
*Amirkabir University & University of Toronto*
*shirali@aut.ac.ir*

*Gerald Penn*
*University of Toronto*
*gpenn@cs.toronto.edu*

## Abstract

Text-To-Speech (TTS) systems are commonly evaluated along two main dimensions: intelligibility and naturalness. While there are clear proxies for intelligibility measurements such as transcription Word-Error-Rate (WER), naturalness is not nearly so well defined. In this paper, we present the results of our attempt to learn what aspects human listeners consider when they are asked to evaluate the "naturalness" of TTS systems. We conducted a user study similar to common TTS evaluations and at the end asked the subject to define the sense of naturalness that they had used. Then we coded their answers and statistically analysed the distribution of codes to create a list of aspects that users consider as part of naturalness. We can now provide a list of suggested replacement questions to use instead of a single oblique notion of naturalness.

**Index Terms**: Text-To-Speech, Naturalness, Evaluation

## 1. Introduction

An important aspect of evaluating Text-to-Speech (TTS) systems nowadays is their naturalness. Recent advances in neural TTS systems have resulted in systems such as Tacotron [1] that are almost indistinguishable from real human voices. The way that such claims are represented through evaluations is by reporting Mean-Opinion-Score (MOS) results of almost 5 on a 1-5 scale. Mean opinion scores for TTS naturalness are a subjective assessment by a human listener as to how natural a sample sounds. Prompts for this task are generally one sentence long and selected from sources such as books.

Naturalness would at one time have been one dimension of a broader TTS evaluation that also includes an intelligibility assessment, such as through transcription error rates of Semantically Uninterpretable Sentences (SUS). TTS systems are now usually compared with each other and/or real human voices. As a result, naturalness is now regarded as an ordinal dimension of speech quality in its own right.

While the definition of a transcription task and its evaluation criteria (Word-Error-Rate or WER) are fairly clear and precise, naturalness is not so well defined. Usually, no definition of what 'natural' means is given to subjects – perhaps out of concern for priming them – nor is there any provision of context within which the sample occurs [2].

What is interesting, however, is that up until about 1995, "natural speech" was the preferred technical term for describing human-generated speech. There was no discussion of an abstract naturalness that synthesizers could approximate on a scale from 1 to 5. There was a very detailed discussion, on the other hand, about the quality of synthesized speech, and indeed the earliest ITU-T P.85 standard [3] for evaluating speech synthesizers was equipped with three so-called Q-type scales

that were designed to measure just that. The first mention of naturalness that we can find was actually in the speech coding literature [4], where it was used to characterise degradations in subjective quality and speaker recognizability that did not also affect intelligibility.

The earliest Blizzard challenges [5] faithfully measured naturalness, along with another feature called similarity, in a context in which every synthesized prompt could be compared to a gold-standard recording of the same prompt by the same voice on which the synthesizer itself had been trained, and so every synthesized sample could be interpreted as an approximation of a human-generated sample. The connection to speech coding was very clear.

A recent [6] comparison of the naturalness of TTS systems and human-generated voices shows that, by the empirical standards of the present-day TTS research community, TTS systems had reached statistical parity with human speech in its degree of naturalness at some point prior to 2013. This forces one to conclude that either the more recent quest for human-like speech quality by deep-learning researchers is simply moot or that the concept of abstract naturalness is not well-founded. Here, we explore better foundations for naturalness.

Another result of that study was that users rank accented speech as less natural, in agreement with older studies [7] that had reported similarities between degradation due to synthesized speech and degradation due to foreign-accented speech, as observed through a dimensionality reduction of more ecologically valid performance measures in the context of speech interfaces for pilot's cockpits by the United States Air Force.

In earlier Blizzard challenges (as per the recommendations for the ITU-Q scales), it was not uncommon to find considerably longer prompts, with very vertically directed instructions on how to establish one's impression:

*"Overall impression: Please try to imagine what your reaction would be if this were an actual telephone message from a mail order house or a request for information from a travel agency."*

*"Acceptance: Please indicate whether or not you find that the voice you heard would be acceptable for such an automatic answering service by telephone."*

These are not precisely defined instructions or definitions, however, as they require introspection on the part of the listener. This is in contrast to the transcription tasks for measuring intelligibility, in which the listener's accuracy is objectively measured.

In this paper, we aim to elucidate an ambiguity in the present naturalness evaluations of TTS systems. We attempt to document how subjects who participate in TTS evaluations will define naturalness for themselves when left to their own

devices. We did this through a user study that mimics the usual evaluation of TTS systems and at the end, explicitly asks the participants to define their notion of naturalness (Section 2). Then, we coded the answers and extracted concepts from them using grounded theory [8] (Section 3). We calculated pointwise mutual information (PMI) scores [9] to identify implicit relations between different aspects of naturalness (Section 4). We report our observations from the coded data in Section 5. This includes identifying some potential problems that relate to the definition of naturalness. We then propose a series of alternative measurements to resolve those problems (Section 6).

The most similar previous work to the present study was an elicitation of dimensions of "quality," together with factor analysis, of roughly 15 German TTS systems in 2011 [10] (see also their excellent literature review of proposed dimensions of evaluation [11]). This study differs from ours in that: (1) their coding was guided primarily by speech experts at Deutsche Telekom, and (2) theirs was not an attempt to characterize naturalness specifically. In fact, three of their attributes were named as "natural," and all three were the strongest correlates to one of three principal components that they determined from their factor analysis.

# 2. Data Collection

As mentioned earlier, it is common in TTS evaluation to measure the naturalness of generated speech. Subjects are asked to express how natural an example prompt is. For example, in the Blizzard challenge, the subject is asked:

*"Now choose a score for how natural or unnatural the sentence sounded. The scale is from 1 [Completely Unnatural] to 5 [Completely Natural]."*

The assumption is that the user already embodies a definition of "natural." Here, we designed a user study that closely mimics the usual TTS evaluations studies, such as the Blizzard challenge.[1]

## 2.1. User Study Structure

Our user study had 5 main parts, the first of which is a standard consent declaration that provides an overview of the study. The second is a demographic questionnaire that collects information similar to what is collected in a Blizzard challenge, such as their age range, whether they are native English speakers, how they would rate their English reading/listening/speaking/writing ability, etc.

### 2.1.1. Individual-prompt naturalness

We ask subjects to perform two types of naturalness assessment. In the first type, they should assess the naturalness of a single prompt. This is usually how TTS evaluations such as Blizzard ask a subject to assess a TTS system. Considering that TTS evaluation tasks also ask subjects to transcribe prompts (which is used to measure the intelligibility of the generated voices), we created a combined question for each prompt that first asks the user to transcribe the text, followed by a question that asks them to assess naturalness. We tried to use questions and prompts that closely resemble those that have been used in previous Blizzard challenges and other TTS

evaluations. Here are the instructions that we showed them for the naturalness assessment:

*Now rate how **natural** or **unnatural** the sentence **sounded:***
1. Completely Unnatural
2. Mostly Unnatural
3. In Between Natural and Unnatural
4. Mostly Natural
5. Completely Natural

### 2.1.2. Pairwise Naturalness Comparison

We also added an extra section asking the user to perform pairwise comparisons of naturalness between prompts that are generated by different systems. Relative assessments are known to have much less variance between subjects. For this part, the user could only listen to each prompt once, and should first listen to prompt A. After listening to both prompts, they should compare the naturalness of the two prompts. Here is the instruction that we gave them:

*"Please listen to the following two voices and compare their naturalness. You should ignore the meanings of the sentences and instead concentrate on how natural or unnatural each one sounded. You can listen to each utterance by clicking on the play button beneath it. Note that you can only listen to the utterance once, and you should listen to voice A first."*

We used almost identical wording to refer to naturalness in this part. They should select one of these five options:
1. Voice A is significantly more natural than voice B
2. Voice A is slightly more natural than voice B
3. Their naturalness is similar
4. Voice B is slightly more natural than voice A
5. Voice B is significantly more natural than voice A

### 2.1.3. Naturalness Definition

After completing the naturalness assessment of different prompts from different speakers, we ask our main question as a single post-study questionnaire:

*"Please define the naturalness definition that you used to rank the naturalness of voices in this user study."*

Our goal was to let the user complete the evaluation of the prompts as they would in other TTS evaluations and only then ask them to define naturalness as it pertained to the evaluation.

## 2.2. Speaker Prompts

We included 20 different speakers in our study: 5 professional speakers (one from the original training data of Blizzard 2013 and 4 other professional speakers), 5 native Indian speakers, 5 native (but not professional) North American speakers, and 5 TTS systems from Blizzard 2013 (systems B, C, D, H and K).

For each speaker, we selected two different sentences to eliminate the effect of text on performance. TTS evaluations generally do this. The sentences were selected from the set of the sentences that were used for the Blizzard 2013 challenge.

---

[1] Considering that we have both human and TTS-generated voices in our study, our ethics board did not allow us to tell the participants that they are evaluating TTS generated voices

(which is common in TTS evaluations). Instead, they allowed us to use the phrase "evaluate computer-generated speech."

### 2.3. Participants

We wanted the conditions of our user study to be similar to a Blizzard challenge. Therefore, it was designed as a web-based study that could be completed over the internet. We recruited participants from Amazon Mechanical Turk [12]. 175 participants completed the study. Amazon Mechanical Turk has been used in various Blizzard challenges and also by different researchers for evaluating TTS systems.

# 3. Coding

### 3.1. Codes

We used grounded theory and the emergent coding approach that is presented in chapter 11 of [8] for coding. Grounded theory is a systematic method that is widely used in the social sciences to develop a basic vocabulary that is "grounded" in data that have been collected pre-theoretically. The goal is to identify and define codes, categories, and finally theories from the data. Two different approaches can be used to identify and define the codes, which are later on combined to define categories: open coding (also known as emergent coding) and selective coding. In open coding, the process starts by extracting potential codes from the data by analyzing the raw data word-by-word, line-by-line and refining them in an iterative process. Selective coding is used once we already have a list of potential codes (e.g., from a previous study) to then identify the ones that are present in the data. Here, we used emergent coding to identify the novel terms that participants use to define naturalness.

We started with one pass in which definitions are analyzed, extracting codewords from them. Each time we saw a new keyword in an answer, we added it to the code list and considered it for the remaining answers. The output of this phase was 146 codewords, while each answer had on average 4.85 keywords.

Most of these codewords only appear in a few answers. For example, more than 85% of them appeared in less than 10 answers, two thirds of them appeared in less than 5, and over one third only appeared in a single answer.

Then, we started to combine codewords that were closely related to each other or were used in the same context with the same meaning. For example, we grouped codewords *Tell* and *Express* together because both describe the same action. Another example is the grouping of codewords *Human*, *People*, *Person*, *Mind*, *Everyone*, and *Someone* that were used to refer to a human speaking. This pass reduced the number of codewords to 39 codes, with an average of 4.61 codes/answer.

### 3.2. Concepts

After finalizing the set of codes, we grouped similar and related codes into concepts [8]. We performed multiple iterations of grouping to finally come up with five concepts. The concepts and related codes are presented in Table 1, along with the number of answers that have that code.

#### 3.2.1. Speech

The first concept consists of codes that describe properties of speech. Half of all answers (88) had at least one of these codes. This shows that for at least half of the subjects, naturalness does in fact relate to the correlates of speech that we as speech researchers seek to measure.

In this concept, the main codes were *Clarity* (34%), *Understand* (27%), *Accent* (23%) and *Tone* (16%). This shows that subjects usually attempt to focus on the clarity of the voice and intelligibility. Because subjects may not have identified transcription as a test of intelligibility, it is possible that the appearance of intelligibility here does not reflect how subjects would define naturalness in a setting where they understood that it was meant to be complementary to transcription error.

Another important code here is *Accent*. 11% of answers have a word that expresses this code (words such as Native, American, Indian, and Foreign).

Table 1: *Concepts and Codes.*

| Concept | Code | Answer Count |
|---|---|---|
| Speech | Accent | 20 |
| | Clarity | 30 |
| | Emotion | 4 |
| | Flow | 11 |
| | Noise | 5 |
| | Pause | 11 |
| | Pitch | 3 |
| | Pronunciation | 11 |
| | Tone | 14 |
| | Smoothness | 8 |
| | Speed | 3 |
| | Understand | 24 |
| Typological | Computer | 48 |
| | Everyday | 9 |
| | Generated | 24 |
| | Human | 64 |
| | Mechanical | 10 |
| | Normal | 22 |
| | Reading | 7 |
| | Real | 18 |
| Descriptors | Adjective | 18 |
| | Adverb | 13 |
| Reflective Process | I | 53 |
| | Feel | 4 |
| | How | 15 |
| | Comparison | 18 |
| | Like | 45 |
| | Mean | 14 |
| | Quality | 5 |
| | Rank | 11 |
| | Should | 7 |
| | Whether | 14 |
| Receiving Information | Hear | 23 |
| | Speak | 26 |
| | Speech | 21 |
| | Sounded | 83 |
| | Tell | 4 |
| | Understand | 24 |
| | Voice | 66 |
| | Word | 21 |

There was also a clear disagreement among subjects about whether accent is or is not a part of naturalness. While most subjects think that having an accent does not reduce naturalness (such as saying "*Even if it were foreign, it could still sound natural*" or "*... not including accents which I did not use as a basis.*"), another contingent believe that it does (e.g., equating

naturalness with speaking with an American accent: "*If the person had an American accent, I thought it was more natural than an Indian accent.*") This is in contrast to the less nuanced finding in [6] that people rank speakers with Indian accents as less natural than speakers with North American accents.

### 3.2.2. Typological

The second concept consists of qualitative typological classes that can be used to define naturalness. Here, there is a tacit assumption that there are two classes of speech: natural and unnatural, and so defining naturalness involves enumerating the properties that distinguish them. Two thirds of all answers use at least one of these codes.

In general, these two groups are *Humans* (55%) and *Computers* (42%). In addition to using those nouns, they also used descriptive adjectives to this effect: *Generated* (21%), *Mechanical* (9%) and *Reading* (6%) for computers and *Normal* (19%), *Real* (16%), *Everyday* (8%) for humans.

### 3.2.3. Receiving Information

The third concept consists of words that describe how information is conveyed to the user from the prompt. The majority of answers (85%) have at least one such code. The top codes in this concept are *Sounded* (56%), *Voice* (45%), *Speak* (18%), *Understand* (16%), and *Hear* (16%).

They can be grouped into two sub-concepts: those that related to generation (*Speak*, *Speech*, *Tell*, *Voice*, and *Word*) and those that related to perception (*Hear*, *Sounded*, and *Understand*).

### 3.2.4. Reflecting Process

The fourth concept is the group of words that convey a process of reflection on the definition of naturalness. Two thirds of answers have at least one such code. The most common one is *I* (46%) (that is a combined code for words such as *I, my, me, we*, etc.), which shows that subjects are attempting to express what naturalness means to them vs. to someone else. Only a small number of answers (6%) have the code *Should*, reflecting an attempt at a normative, global definition of naturalness.

Other common codes in this concept are *Like* (39%), *Compare* (15%), and *Whether* (12%). They are used alongside typological concept codes to express that many subjects consider a rating by naturalness to be tantamount to finding the typological class (human or computer) that the prompt belongs to. This again reflects that 'natural' is equivalent to human-generated and 'unnatural,' to computer-generated. For example, one subject says "*Naturalness to me means something that comes out of the person.*"

### 3.2.5. Descriptors

The fifth concept was adjectives and adverbs that they use to better express their idea. For example, they may say "understand easily". 17% of answers have such a descriptor.

## 4. PMI

One of the most important lexical-semantic conventions in NLP is the use of Pointwise Mutual Information (PMI) [9], because of its association with commonly two words appear together. The PMI of two words $x$ and $y$ is defined as:

$$PMI(x,y) = log_2 \frac{p(x,y)}{p(x)p(y)}$$

In this formula, $p(x)$ is the probability of $x$ appearing in the text, $p(y)$ is the probability of $y$ appearing in the text, and $p(x,y)$ is the probability of both $x$ and $y$ appearing in a text.

Here, we calculated probability as relative frequency to the total number of answers (175). Then we calculated PMI for all code pairs. The results are shown in Figure 1. In this figure, only the PMI for pairs that appeared together (i.e., $p(x,y)$>0) is shown. Color coding is used to give a visual overview of scores: red cells are negative values, yellow cells are between 0 and 1, and green cells are above 1. Note that only the lower half of the matrix is shown because of the symmetry of the PMI score.

Figure 2 shows only pairs with PMI>1. It also separates the 5 concepts from each other.

The first observation that we can make from Figure 2 is the relatively high co-occurrence of codes within the speech concept. For example, the code *Speed* has high PMI with codes such as *Pause*, *Pitch*, *Pronunciation*, and *Tone* (all of them >2 and PMI(*Speed*,*Pitch*)=4.281 which is the highest PMI score in our data). The second highest PMI is PMI(*Pitch*, *Emotion*) = 3.866. This is also expected.

The *Pause* code also has high PMI with *Tone* (2.506) and *Smoothness* (2.577). The latter is interesting because it may serve as a partial definition of smoothness. There is also a high PMI between *Pitch* and *Tone* (3.059) which is expected.

Some of the speech codes also have high PMI with reflecting-process codes. For example, we have PMI(*Feel*,*Noise*) = 3.129 and PMI(*Quality*, *Speed*)=3.544. The first one is interesting in that people feel the need to subjectively qualify their perception of Noise.

The *Tell* code also has high PMI with a few other codes: PMI(*Tell*, *Reading*) = 3.644, PMI(*Tell*, *Feel*) = 3.451, and PMI(*Tell*, *Mean*) = 2.644.

The *Mechanical* code has high PMI with *Pause* (2.255), *Tone* (1.907), and *Noise* (1.807), which again serves to interpret the term. For example, this is the definition that one participant gave (the word *Electronic* instantiates the *Mechanical* code):

"*I decided both on how normal the rythym* [sic] *of the speech sounded. In other words were the pauses and flow consistant* [sic] *with normal speech. Also though how soft or smooth it was as compared to a less smooth electronic sounding voice.*"

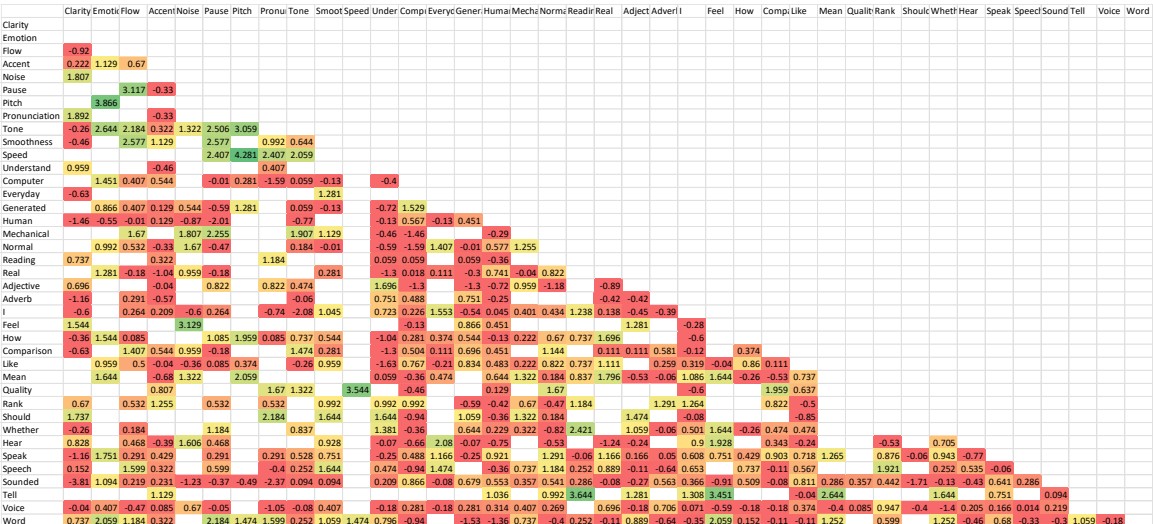

Figure 1: *PMI of code pairs.*

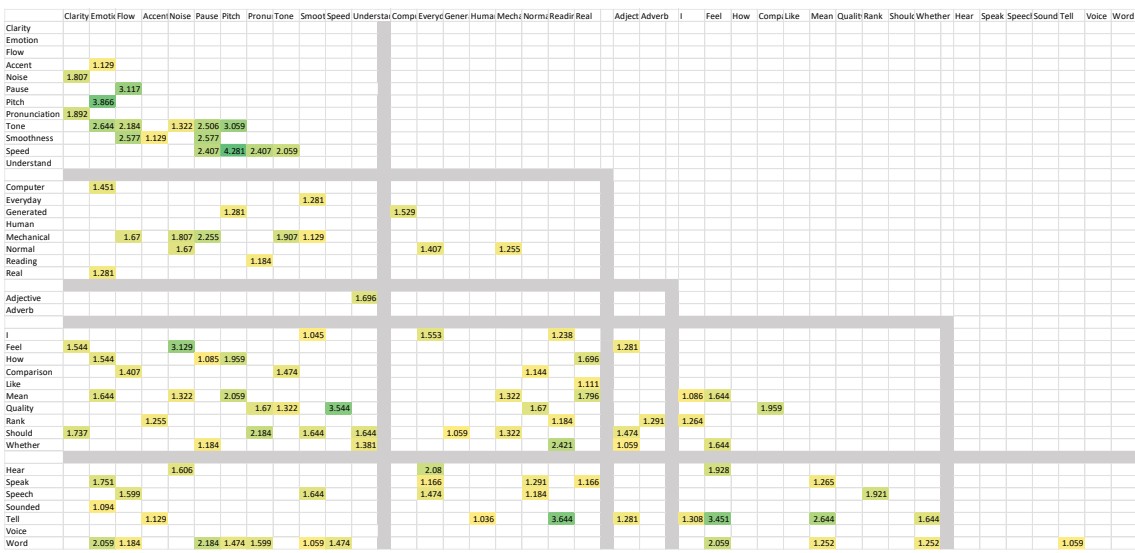

Figure 2: *Code pairs with PMI>1.*

# 5.  Analysis

While PMI provides an overview of how different codes appear together in an answer, it does not give a complete overview of how people define naturalness with these codes. In this section, we provide the results of our deeper inspection of answers by way of a few important, summarial points.

## 5.1.  A priori definition of Naturalness

As we mentioned earlier, it is common among speech researchers to assume that there is an *a priori* definition of naturalness. We have observed multiple instances of participants assuming the same thing, who therefore believe that there is no need to define naturalness. Some of them seem to consider naturalness to be a primitive quality that defies definition. Here are two example answers:

*"I'm not sure what you are asking, we were asked to rank which voices sounded more natural and less like computer-generated voices."*

*"What? if it sounded more natural I voted it to sound more natural. what is this question asking?"*

Our other observations reveal, however, that the view of naturalness as an agreed-upon or inscrutable primitive cannot be completely correct.

## 5.2.  Lack of confidence in defining naturalness

We also observed a lack of confidence in some participants about the existence of a universally accepted definition of naturalness. The use of codes from the reflecting-process concept shows that some subjects express *how they define naturalness* and not *how it is defined for everyone*. This suggests that we could remove ambiguity by providing a clear and concise characterization of naturalness (or as we will discuss shortly, using multiple substitution to cover different aspects of it).

Another cause of this perceived lack of universal definition could be the difference between people in assigning numerical (or level-based) scores to prompts. This can be referred to as the

*normalization problem.* One way of preventing this error is to ask subjects to compare the naturalness of two prompts together (rather than asking them for individual numerical ranks).

### 5.3. Inconsistent and contradicting definitions

We also noticed that sometimes subjects have contradictory opinions about how different properties of speech affect naturalness. The most prominent example is the speaker's accent. Referring to the accent of the speaker was a common theme, although they used different terms to do it (such as *Accent*, *Native*, *Indian*, *American*, *Foreign*, etc.). And while most of the people who used such terms were trying to say that this factor does NOT affect naturalness, there are examples to the contrary:

*"If the person had an American accent, I thought it was more natural than an Indian accent."*

As mentioned above, previous research [6] has revealed that even if most users think that they are ignoring accent, in practice they would rank speech samples with accent as less natural. This can be a matter of unconscious bias.

Our proposed solution is to ask subjects to rank samples on the basis an explicit subset of criteria related by this study to naturalness. We will discuss this in the next section.

### 5.4. Equity with being human

We noticed that subjects usually equate naturalness with being generated by a human (i.e., being human-generated at its source, even if recorded). This is even consistent with the prevailing technical use of "natural speech" among speech researchers until the 1990s. Although subjects referred to aspects such as understandability and clarity, their main criteria for assessing naturalness seem to be cues that the true generative source of the prompt as human or artificial.

This is an important point to consider. If people equate naturalness with humanity, then no computer-generated voice can be completely natural other than through deception. It is unclear whether questions about naturalness could ever serve their originally intended purpose if viewed through the lens of discovering a potential deception. This would imply, for example, that informing human subjects that a prompt had been computer-generated would bias them against assigning a high naturalness score.

### 5.5. Measuring intelligibility as a part of naturalness

The last observation that we want to report is how people used aspects such as *Clarity* and *Understand* to characterize naturalness. These concepts, especially understandability, are usually considered to be part of intelligibility tests such as transcription tasks. This suggests that the usual partitioning of tests into intelligibility and naturalness may not be completely well-founded. It is unclear, for example, whether having a separate test of intelligibility would completely remove intelligibility from consideration in an assessment of naturalness.

## 6. Proposed Alternatives

In the previous section, we presented some problems with current approaches to measuring naturalness. To solve them, we propose a set of alternative questions to use instead of a single question about the naturalness of a prompt. These alternative questions can be grouped into two sets: speech properties and human-similarity.

### 6.1. Measuring Speech Properties

The first set consists of five aspects of speech properties to consider. They include *Clarity*, *Understandability*, *Fluency*, *Pronunciation*, and *Accent*. We selected these aspects based on the main speech properties that subjects mentioned in their definitions, including the codes with highest repetition. We did not distinguish between *Fluency* and *Pauses* because their differences may not be clear to many users. Furthermore, some aspects such as *Tone* and *Pitch* were not used very consistently across subjects. We can use either a 5-point or 3-point Likert-scale question for each. For example, for the *Clarity* aspect, we can ask the following question:

*"Please rate the clarity of the voice in each prompt, on a scale of 1 (completely unclear) to 5 (perfectly clear)"*

The question can also be framed as agreement by the user. For example, we can instead ask:

*"Please describe whether you agree or disagree with the following statement (1 means strongly disagree, 5 means strongly agrees): The speaker speaks clearly."*

The benefit of the second type is that we can use the question and potential answer format for all 5 aspects (clarity, understandability, fluency, pronunciation, and accent). It also focusses attention on the user's own personal assessments, rather than on considerations of universality.

### 6.2. Measuring the Human-Similarity

The second set of questions focusses on whether a prompt is human-generated. We propose the following three questions:

1. Please select whether the voice sounds more like a human or a computer (mostly like a computer, a bit like a computer, in-between a computer and a human, a bit like a human, mostly like a human).

2. Please select how mechanical or natural the voice sounds (mostly mechanical, a bit mechanical, neither mechanical nor normal, a bit natural, mostly natural).

3. Determine whether the voice sounds like it is being read from text or being spoken spontaneously during a conversation (more like being read from a written text, partly being read from a text and partly being said spontaneously, mostly said spontaneously).

These questions are essentially asking the same thing, but at three different levels: first to conclusively resolve any would-be deception as to human vs. computer; second by attending to the main property associated with these types of voices (note that 'natural' here is narrowly used with the meaning, 'not mechanical'); then third by resolving the second-greatest and still remaining typological consideration: scripted vs. spontaneous. This third consideration was clearly visible in some of the definitions that users provided, such as:

*"Naturalness to me means something that comes out of the person. He's not reading something, he's saying something that is coming out of his mind."*

## 7. Conclusion and Future Work

In this paper, we revisited the problem of what naturalness means when evaluating TTS systems. We argued that the current approach of measuring naturalness simply by asking

about it is not sufficiently well defined. We conducted a study that asked to subjects to perform fairly usual TTS evaluations and at the end ask what naturalness definition they used. We coded the answers and grouped them into concepts.

We performed a co-occurrence analysis and also an in-depth textual analysis of the answers and presented our main observations: an assumption that there is no need to define naturalness at all, alongside the paradoxical lack of a universal definition for it, contradictory aspects in the definition of naturalness, the interpretation as human vs. computer, and overlap with intelligibility.

We presented a set of alternative questions and measurements to use instead of a single question to rank naturalness. The main takeaway of our paper for TTS researchers is that there is a need to clearly define naturalness for participants, and furthermore to clearly distinguish its different aspects during evaluation.

We plan to conduct a follow-up study that uses these new questions alongside the existing practice of overall naturalness, with and without intelligibility assessments.

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
