# OpenReview forum: "Better Replacement for TTS Naturalness Evaluation"
_Interspeech.org/2023/Workshop/SSW — SSW12_

### Official Review · Reviewer_3nne · 2023-06-02
**Qualitative methods for analyzing free-text definitions of naturalness from listeners**

**Rating:** 4
**Confidence:** 1

**Review:**

This paper uses qualitative grounded-theory based methods to analyze free-text explanations by listeners about how they decided on naturalness ratings after completing a listening test with MOS and pairwise comparisons.  To be honest, I come from a more quantitative background and I was not at all familiar with grounded theory before reading this paper, and so I had to do a bit of background reading to familiarize myself.  Thus, I don't think that I am the most well-equipped person to evaluate the authors' specific *application* of grounded theory in this paper (hence my "low confidence" choice).  Nevertheless, I think that I am more or less representative of at least a certain subset of SSW readers, and so this review has several suggestions for making the paper more accessible for this audience.

It was exciting to see some work being done on actually asking listeners about their reasons for rating samples as natural or unnatural.  Furthermore, from what I understood of grounded theory, it seems like a good and appropriate choice for analyzing this kind of free-text subjective data.

However, the main difficulty I had in understanding this paper was with the description in Section 3 about how exactly the coding was conducted.  Chapter 11 of Reference [8] is cited, but the paper needs to stand on its own and have at least a brief description of the theory and the coding approach. Furthermore, even after reading up on grounded theory, I still could not understand the following points, which I think are crucial for understanding the paper:

* "one pass of analyzing all definitions" -- which definitions?
* "each time we saw a new keyword in an answer" -- how were words identified as keywords?
* What are codewords vs. keywords?
* How exactly are codewords combined?  What criteria is used to decide that they are related?  In particular, in reading about grounded theory, I learned that prior to coding, an annotation scheme is typically defined in order to ensure that coding is done in a consistent and verifiable manner.  Perhaps more details about the coding scheme would be helpful.  Also, there was no information provided about how many coders labeled the data, or what their agreement was.
* After grouping the codes, another grouping is conducted to group codes into concepts.  How is this different from the previous grouping?
* Some codes, e.g. "Understand," appear under two different concepts ("Speech" and "Receiving Information").  How did this happen, what does it mean, and how was this handled in analysis?
* How was word sense handled?  E.g., "Mean" can refer to an intended meaning, or an average, or unkindness; "Like" can refer to "similar" or to having a preference for something.  I am assuming that word sense was taken into account somehow as per the description "were used in the same context with the same meaning," but a bit more explanation/clarification on this point would be helpful.

Next, the conclusions drawn in the PMI section and elsewhere in the paper are vague, seem to contain leaps of logic that are not supported by the data in a way that is apparent to the reader, or are otherwise difficult to understand.  For example:

* "The latter is interesting because it may serve to define smoothness."  How so / what does this mean?
* "The first one is interesting in that people feel the need to subjectively qualify their perception of Noise."  I am not sure that this claim is necessarily or clearly supported by the fact that the words Feel and Noise co-occur frequently.
* The "Tell" code having high PMI with several other codes: what can we learn from this?
* "We did not distinguish between *Fluency* and *Pauses* because their differences may not be clear to many users."  How was this determined / what was this based on?

There were also some smaller claims and points about TTS, listening tests in general, and the specific listening test that was conducted that were questionable or needed clarification:

* "Usually, a TTS evaluation protocol has two parts: naturalness measurement via Mean-Opinion-Scores (MOS) and intelligibility measurement via transcription error rates of Semantically Uninterpretable Sentences (SUS)."  This is a minor point, but I am not sure about the "usually" here -- these days, I see very few TTS papers evaluating intelligibility using SUS, and many papers evaluating for other factors like speaker similarity.
* "so-called Q-type scales" -- what are these exactly, and how do they relate to other types of rating methods?
* "along with another feature called similarity" -- this is in fact speaker similarity (to the target speaker) which would be good to clarify.
* Why were listeners only allowed to listen to each sample once in the pairwise test?  I believe that this is standard for intelligibility tests, to prevent listeners from listening to less-intelligible sentences enough times to be able to catch all of the words eventually, but not for pairwise comparisons.  Was this same restriction imposed on the MOS task?
* In Section 2.2, there is different information about each speaker in terms of their language and whether or not they are a professional speaker -- it would be better to include all of these facets of information for all of the speakers.
* How and why were the 5 TTS systems from Blizzard 2013 chosen?  This is especially relevant because the types of TTS systems that were chosen may affect the way that listeners think about naturalness and respond to the survey question -- the *types of artifacts and unnaturalness that they are exposed to during the test* would be expected to inform their answers.
* Were there any requirements imposed on the listeners recruited from MTurk?  e.g., language, geographic location, or a percent of previously-accepted HITs?  Were any participants excluded for not providing good answers, such as answers that were basically empty, or not answering the question?
* "This suggests that the usual partitioning of tests into intelligibility and naturalness may not be completely well-founded."  Indeed, it's well-known that naturalness and intelligibility are correlated -- see the second reference listed later.

It was really nice to see the exact text presented to listeners, though -- this is too often missing from papers that report listening test results.

There were several sentences and phrases throughout the paper whose meaning was generally unclear:

* "... which again serves to interpret the term.  For example, this is the definition that a participant given that refers to this"
* "second by attending to the main property associated with these types of voices"
* "very vertically directed instructions"

-- I could not understand the meaning of these sentences/phrases.

There were also several issues related to presentation and typography:

* The order of the concepts in Table 1 and the order in which they are presented in the text is different.
* Figure 1 contains a lot of information, but it is not clear at all what should be learned from it (and there is no explanation for this in the text).

Minor typos:

* pilot's cockpits  -> pilots' cockpits
* User are asked -> Users are asked

The works cited seemed a bit light.  Several papers from Sebastian Möller's lab come to mind as being relevant, for example:

* https://www.isca-speech.org/archive_v0/ssw8/papers/ssw8_147.pdf  which was a meta-study of several past listening tests from which 5 main perceptual factors relating to synthesized speech quality were derived; and
* https://www.essv.de/pdf/2015_105_111.pdf  which is prior work showing that naturalness and intelligibility are in fact correlated.

Overall, it was refreshing to see work that actually asks listeners for their reasons / internal definition of naturalness.  I think that there is generally plenty of room for this type of qualitative analysis of listener opinions, and this is a promising line of inquiry.  This paper also presented several interesting findings, such as the fact that listeners were very split on whether they considered accented speech to be natural or not.  The suggestions for alternate questions other than about "naturalness" to ask, based on the codes with the highest repetition, is also a nice contribution.  I think the paper as it is written now has too many unclear points regarding the methodology and tenuous connections between the conclusions drawn and the evidence for them, but I would nevertheless encourage the authors to continue pursuing and developing this work.

---

### Official Review · Reviewer_sBhZ · 2023-06-06
**A useful, but an incomplete effort to describe an alternate to MOS**

**Rating:** 5
**Confidence:** 4

**Review:**

This paper describes a critique of Mean-Opinion Score for evaluation of TTS naturalness as well as a proposal for evaluating naturalness.

The major weakness of this paper is that is omits a lot of prior work around subjective assessment that informs the use of MOS and previous critiques e.g. "Mean Opinion Score (MOS) revisited: Methods and applications, limitations and alternatives" by Streijl, Winkler and Hands https://stefan.winklerbros.net/Publications/mmsj2016.pdf and "Measuring the naturalness of synthetic speech" by Nusbaum, Francis and Henly https://link.springer.com/article/10.1007/BF02277176.  The use of MOS asking about "naturalness" comes from an acknowledgement that "naturalness" is a multifaceted phenomenon, but people are able to synthesize these and provide a useful and actionable response to a single question, simplifying the assessment.

That said, the existence of previous critiques, does not negate the value of the criticism offered in this paper or a fresh consideration of alternate forms of evaluation.  The quality of speech synthesis has advanced dramatically since this early to mid 1990s when MOS emerged as a de-facto standard for synthesis naturalness assessment.

This project to improve evaluation of naturalness seems incomplete.  The critique of MOS evaluation is well described.  The technique to understand what subjects are responding to when they assess naturalness is reasonable, as is the corresponding analysis.    However, the use of these findings as a mechanism to improve MOS is under developed.  There are some proposals for measuring specific speech properties and human-similarity, but these have not yet been evaluated on human raters.  It is necessary to understand how these proposed assessments impact evaluation (do raters agree more with respect to these questions than MOS? is it faster to rate systems using these scores of MOS? Are these ratings more statistically powerful to distinguish systems, demonstrating less bias per rater and utterance?). Without this component, the promise of a "Better Replacement for TTS Naturalness Evaluation" is unfulfilled.

Detail:
The tables in Figures 1 and 2 are not readable.  It would be useful to distill the most informative findings from these tables rather than including all of the data in the paper.

---

### Decision · Program_Chairs · 2023-06-14

**Decision:**

Accept

**Comment:**

SSW2003 received 45 papers. The acceptance rate is 82%. We are pleased to inform you that your paper has been accepted by the SSW2023 Program Committee. Please read the reviews carefully and submit your camera-ready paper by June 28th. Most of reviewers performed a detailed review. Please answer to their questions and take into account their comments.
Since your paper received a score below 5/9 that is strongly argued by the reviewers, note that the Program Committee will check if your manuscript has been significantly changed to specifically consider their remarks. Note that camera-ready papers are credited with one extra page to allow authors to consider reviewers’ suggestions. So max 7 pages in total including figures & refs.
The deadline for submitting the revised version (with full non anonymized authors and refs!) is 28th June.